# MUSIC TRANSFORMER:
# GENERATING MUSIC WITH LONG-TERM STRUCTURE

**Cheng-Zhi Anna Huang**[*]  **Ashish Vaswani**   **Jakob Uszkoreit**   **Noam Shazeer**
**Ian Simon**   **Curtis Hawthorne**   **Andrew M. Dai**   **Matthew D. Hoffman**
**Monica Dinculescu**   **Douglas Eck**
Google Brain
{annahuang,avaswani,usz,noam,
 iansimon,fjord,adai,mhoffman,noms,deck}@google.com

## ABSTRACT

Music relies heavily on repetition to build structure and meaning. Self-reference occurs on multiple timescales, from motifs to phrases to reusing of entire sections of music, such as in pieces with ABA structure. The Transformer (Vaswani et al., 2017), a sequence model based on self-attention, has achieved compelling results in many generation tasks that require maintaining long-range coherence. This suggests that self-attention might also be well-suited to modeling music. In musical composition and performance, however, relative timing is critically important. Existing approaches for representing relative positional information in the Transformer modulate attention based on pairwise distance (Shaw et al., 2018). This is impractical for long sequences such as musical compositions because their memory complexity for intermediate relative information is quadratic in the sequence length. We propose an algorithm that reduces their intermediate memory requirement to linear in the sequence length. This enables us to demonstrate that a Transformer with our modified relative attention mechanism can generate minute-long compositions (thousands of steps) with compelling structure, generate continuations that coherently elaborate on a given motif, and in a seq2seq setup generate accompaniments conditioned on melodies[1]. We evaluate the Transformer with our relative attention mechanism on two datasets, JSB Chorales and Maestro, and obtain state-of-the-art results on the latter.

## 1   INTRODUCTION

A musical piece often consists of recurring elements at various levels, from motifs to phrases to sections such as verse-chorus. To generate a coherent piece, a model needs to reference elements that came before, sometimes in the distant past, and then repeat, vary, and further develop them to create contrast and surprise. Intuitively, self-attention (Parikh et al., 2016) could be a good match for this task. Self-attention over its own previous outputs allows an autoregressive model to access any part of the previously generated output at every step of generation. By contrast, recurrent neural networks have to learn to proactively store elements to be referenced in a fixed size state or memory, making training potentially much more difficult. We believe that repeating self-attention in multiple, successive layers of a Transformer decoder (Vaswani et al., 2017) can help capture the multiple levels at which self-referential phenomena exist in music.

In its original formulation, the Transformer relies on absolute position representations, using either positional sinusoids or learned position embeddings that are added to the per-position input representations. Recurrent and convolutional neural networks instead model position in relative terms: RNNs through their recurrence over the positions in their input, and CNNs by applying kernels that effectively choose which parameters to apply based on the relative position of the covered input representations.

---

[*]Google AI Resident. Correspondence to: Cheng-Zhi Anna Huang <annahuang@google.com>
[1]Samples are available for listening at
https://goo.gl/magenta/music-transformer-examples.
Later blog post at https://magenta.tensorflow.org/music-transformer

Music has multiple dimensions along which relative differences arguably matter more than their absolute values; the two most prominent are timing and pitch. To capture such pairwise relations between representations, Shaw et al. (2018) introduce a relation-aware version of self-attention which they use successfully to modulate self-attention by the distance between two positions. We extend this approach to capture relative timing and optionally also pitch, which yields improvement in both sample quality and perplexity for the JSB Chorales dataset. As opposed to the original Transformer, samples from a Transformer with our relative attention mechanism maintain the regular timing grid present in this dataset. The model furthermore captures global timing, giving rise to regular phrases.

The original formulation of relative attention (Shaw et al., 2018) requires $O(L^2D)$ memory where $L$ is the sequence length and $D$ is the dimension of the model's hidden state. This is prohibitive for long sequences such as those found in the Maestro dataset of human-performed virtuosic, classical piano music (Hawthorne et al., 2019). In Section 3.4, we show how to reduce the memory requirements to $O(LD)$, making it practical to apply relative attention to long sequences.

The Maestro dataset consists of MIDI recorded from performances of competition participants, bearing expressive dynamics and timing on a less than 10-millisecond granularity. Discretizing time in a fixed grid on such a resolution would yield unnecessarily long sequences as not all events change on the same timescale. We hence adopt a sparse, MIDI-like, event-based representation from (Oore et al., 2018), allowing a minute of music with a 10-millisecond resolution to be represented at lengths around 2K. This is in contrast to a 6K to 18K length that would be needed on a serialized multi-attribute fixed-grid representation. As position in sequence no longer corresponds to time, a priori it is not obvious that relative attention should work as well with such a representation. However, we will show in Section 4.2 that it does improve perplexity and sample quality over strong baselines.

We speculate that idiomatic piano gestures such as scales, arpeggios and other motifs all exhibit a certain grammar and recur periodically, hence knowing their relative positional distances makes it easier to model this regularity. This inductive bias towards learning relational information, as opposed to patterns based on absolute position, suggests that the Transformer with relative attention could generalize beyond the lengths it was trained on, which our experiments in Section 4.2.1 confirm.

## 1.1 CONTRIBUTIONS

**Domain contributions** We show the first successful use of Transformers in generating music that exhibits long-term structure. Before our work, LSTMs were used at timescales of 15s (~500 tokens) of piano performances (Oore et al., 2018). Our work demonstrates that Transformers not only achieve state-of-the-art perplexity on modeling these complex expressive piano performances, but can also generate them at the scale of minutes (thousands of tokens) with remarkable internal consistency. Our relative self-attention formulation is essential to the model's quality. In listening tests (see Section 4.2.3), samples from models with relative self-attention were perceived as more coherent than the baseline Transformer model (Vaswani et al., 2017). Relative attention not only enables Transformers to generate continuations that elaborate on a given motif, but also to generalize and generate in consistent fashion beyond the length it was trained on (see Section 4.2.1). In a seq2seq setup, Transformers can generate accompaniments conditioned on melodies, enabling users to interact with the model.

**Algorithmic contributions** The space complexity of the relative self-attention mechanism in its original formulation (Shaw et al., 2018) made it infeasible to train on sequences of sufficient length to capture long-range structure in longer musical compositions. To address this, we present a crucial algorithmic improvement to the relative self-attention mechanism, dramatically reducing its memory requirements from $O(L^2D)$ to $O(LD)$. For example, the memory consumption per layer is reduced from 8.5 GB to 4.2 MB (per head from 1.1 GB to 0.52 MB) for a sequence of length $L = 2048$ and hidden-state size $D = 512$ (per head $D_h = \frac{D}{H} = 64$, where number of heads is $H = 8$) (see Table 1), allowing us to use GPUs to train the relative self-attention Transformer on long sequences.

## 2 RELATED WORK

Sequence models have been the canonical choice for modeling music, from Hidden Markov Models to RNNs and Long Short Term Memory networks (e.g., Eck & Schmidhuber, 2002; Liang, 2016; Oore et al., 2018), to bidirectional LSTMs (e.g., Hadjeres et al., 2017). Successful application of

sequential models to polyphonic music often requires serializing the musical score or performance into a single sequence, for example by interleaving different instruments or voices. Alternatively, a 2D pianoroll-like representation (see A.1 for more details) can be decomposed into a sequence of multi-hot pitch vectors, and their joint probability distributions can be captured using Restricted Boltzmann Machines (Smolensky, 1986; Hinton et al., 2006) or Neural Autoregressive Distribution Estimators (NADE; Larochelle & Murray, 2011). Pianorolls are also image-like and can be modeled by CNNs trained either as generative adversarial networks (GAN; Goodfellow et al., 2014). (e.g., Dong et al., 2018) or as orderless NADEs (Uria et al., 2014; 2016) (e.g., Huang et al., 2017).

Lattner et al. (2018) use self-similarity in style-transfer fashion, where the self-similarity structure of a piece serves as a template objective for gradient descent to impose similar repetition structure on an input score. Self-attention can be seen as a generalization of self-similarity; the former maps the input through different projections to queries and keys, and the latter uses the same projection for both.

Dot-product self-attention is the mechanism at the core of the Transformer, and several recent works have focused on applying and improving it for image generation, speech, and summarization (Parmar et al., 2018; Povey et al., 2018; Liu et al., 2018). A key challenge encountered by each of these efforts is scaling attention computationally to long sequences. This is because the time and space complexity of self-attention grows quadratically in the sequence length. For relative self-attention (Shaw et al., 2018) this is particularly problematic as the space complexity also grows linearly in the dimension, or depth, of the per-position representations.

## 3 MODEL

### 3.1 DATA REPRESENTATION

We take a language-modeling approach to training generative models for symbolic music. Hence we represent music as a sequence of discrete tokens, with the vocabulary determined by the dataset. Datasets in different genres call for different ways of serializing polyphonic music into a single stream and also discretizing time.

The JSB Chorale dataset consists of four-part scored choral music, which can be represented as a matrix where rows correspond to voices and columns to time discretized to sixteenth notes. The matrix's entries are integers that denote which pitch is being played. This matrix can than be serialized in raster-scan fashion by first going down the rows and then moving right through the columns (see A.1 for more details). Compared to JSB Chorale, the piano performance data in the Maestro dataset includes expressive timing information at much finer granularity and more voices. For the Maestro dataset we therefore use the performance encoding proposed by Oore et al. (2018) which consists of a vocabulary of 128 NOTE_ON events, 128 NOTE_OFFs, 100 TIME_SHIFTs allowing for expressive timing at 10ms and 32 VELOCITY bins for expressive dynamics (see A.2 for more details).

### 3.2 BACKGROUND: SELF-ATTENTION IN TRANSFORMER

The Transformer decoder is a autoregressive generative model that uses primarily self-attention mechanisms, and learned or sinusoidal position information. Each layer consists of a self-attention sub-layer followed by a feedforward sub-layer.

The attention layer first transforms a sequence of $L$ $D$-dimensional vectors $X = (x_1, x_2, \ldots, x_L)$ into queries $Q = XW^Q$, keys $K = XW^K$, and values $V = XW^V$, where $W^Q$, $W^K$, and $W^V$ are each $D \times D$ square matrices. Each $L \times D$ query, key, and value matrix is then split into $H$ $L \times D_h$ parts or attention heads, indexed by $h$, and with dimension $D_h = \frac{D}{H}$, which allow the model to focus on different parts of the history. The scaled dot-product attention computes a sequence of vector outputs for each head as

$$Z^h = \text{Attention}(Q^h, K^h, V^h) = \text{Softmax}\left(\frac{Q^h K^{h\top}}{\sqrt{D_h}}\right) V^h. \tag{1}$$

The attention outputs for each head are concatenated and linearly transformed to get $Z$, a $L$ by $D$ dimensional matrix. A upper triangular mask ensures that queries cannot attend to keys later in the sequence. For other details of the Transfomer model, such as residual connections and learning rates, the reader can refer Vaswani et al. (2017). The feedforward (FF) sub-layer then takes the output $Z$

from the previous attention sub-layer, and performs two layers of point-wise dense layers on the depth $D$ dimension, as shown in Equation 2. $W_1, W_2, b_1, b_2$ are weights and biases of those two layers.

$$FF(Z) = \text{ReLU}(ZW_1 + b_1)W_2 + b_2 \tag{2}$$

### 3.3 RELATIVE POSITIONAL SELF-ATTENTION

As the Transformer model relies solely on positional sinusoids to represent timing information, Shaw et al. (2018) introduced relative position representations to allow attention to be informed by how far two positions are apart in a sequence. This involves learning a separate relative position embedding $E^r$ of shape $(H, L, D_h)$, which has an embedding for each possible pairwise distance $r = j_k - i_q$ between a query and key in position $i_q$ and $j_k$ respectively. The embeddings are ordered from distance $-L + 1$ to 0, and are learned separately for each head. In Shaw et al. (2018), the relative embeddings interact with queries and give rise to a $S^{rel}$, an $L \times L$ dimensional logits matrix which modulates the attention probabilities for each head as:

$$\text{RelativeAttention} = \text{Softmax}\left(\frac{QK^\top + S^{rel}}{\sqrt{D_h}}\right)V. \tag{3}$$

We dropped head indices for clarity. Our work uses the same approach to infuse relative distance information in the attention computation, while significantly improving upon the memory footprint for computing $S^{rel}$. For each head, Shaw et al. (2018) instantiate an intermediate tensor $R$ of shape $(L, L, D_h)$, containing the embeddings that correspond to the relative distances between all keys and queries. $Q$ is then reshaped to an $(L, 1, D_h)$ tensor, and $S^{rel} = QR^\top$.[2] This incurs a total space complexity of $O(L^2D)$, restricting its application to long sequences.

### 3.4 MEMORY EFFICIENT IMPLEMENTATION OF RELATIVE POSITION-BASED ATTENTION

We improve the implementation of relative attention by reducing its intermediate memory requirement from $O(L^2D)$ to $O(LD)$, with example lengths shown in Table 1. We observe that all of the terms we need from $QR^\top$ are already available if we directly multiply $Q$ with $E^r$, the relative position embedding. After we compute $QE^{r\top}$, its $(i_q, r)$ entry contains the dot product of the query in position $i_q$ with the embedding of relative distance $r$. However, each relative logit $(i_q, j_k)$ in the matrix $S^{rel}$ from Equation 3 should be the dot product of the query in position $i_q$ and the embedding of the relative distance $j_k - i_q$, to match up with the indexing in $QK^\top$. We therefore need to "skew" $QE^{r\top}$ so as to move the relative logits to their correct positions, hence $S^{rel} = \text{Skew}(QE^r)$. The "skewing" procedure is illustrated in Figure 1 and will be detailed in the next section. The time complexity for both methods are $O(L^2D)$, while in practice our method is 6x faster at length 650 as prior work still requires manipulating larger tensors.

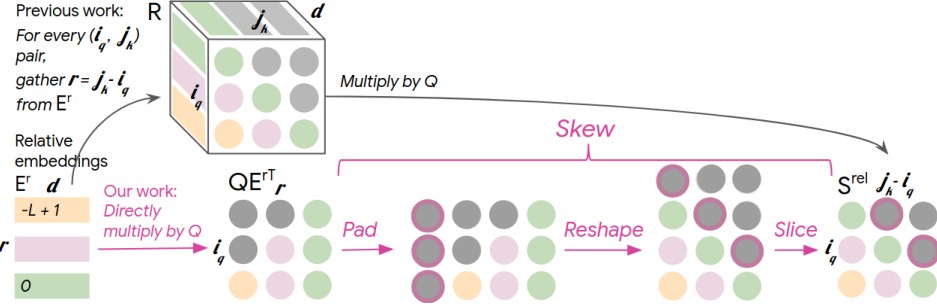

Figure 1: Relative global attention: the bottom row describes our memory-efficient "skewing" algorithm, which does not require instantiating $R$ (top row, which is $O(L^2D)$). Gray indicates masked or padded positions. Each color corresponds to a different relative distance.

---

[2]We assume that the batch size is 1 here. With a batch size of $B$, $Q$ would be reshaped to $(L, B, D_h)$ and $S^{rel}$ would be computed with a batch matrix–matrix product.

Table 1: Comparing the overall relative memory complexity (intermediate relative embeddings ($R$ or $E^r$) + relative logits $S^{rel}$), the maximal training lengths that can fit in a GPU with 16GB memory assuming $D_h = 64$, and the memory usage per layer per head (in MB).

| Implementation | Relative memory | Maximal $L$ | $L = 650$ | $L = 2048$ | $L = 3500$ |
|---|---|---|---|---|---|
| Shaw et al. (2018) | $O(L^2D + L^2)$ | 650 | 108 + 1.7 | 1100 + 16 | 3100 + 49 |
| Ours | $O(LD + L^2)$ | 3500 | 0.17 + 1.7 | 0.52 + 16 | 0.90 + 49 |

### 3.4.1 THE "SKEWING" PROCEDURE

Hence, we propose a "skewing" procedure to transform an absolute-by-relative $(i_q, r)$ indexed matrix into an absolute-by-absolute $(i_q, j_k)$ indexed matrix. The row indices $i_q$ stay the same while the columns indices are shifted according to the following equation: $j_k = r - (L - 1) + i_q$. For example in Figure 1 the upper right green dot in position $(0, 2)$ of $QE^{r\top}$ after skewing has a column index of $2 - (3 - 1) + 0 = 0$, resulting in a position of $(0, 0)$ in $S^{rel}$.

We outline the steps illustrated in Figure 1 below.

1. Pad a dummy column vector of length $L$ before the leftmost column.

2. Reshape the matrix to have shape $(L+1, L)$. (This step assumes NumPy-style row-major ordering.)

3. Slice that matrix to retain only the last $l$ rows and all the columns, resulting in a $(L, L)$ matrix again, but now absolute-by-absolute indexed, which is the $S^{rel}$ that we need.

### 3.5 RELATIVE LOCAL ATTENTION

For very long sequences, the quadratic memory requirement of even baseline Transformer is impractical. Local attention has been used for example in Wikipedia and image generation (Liu et al., 2018; Parmar et al., 2018) by chunking the input sequence into non-overlapping blocks. Each block then attends to itself and the one before, as shown by the smaller thumbnail on the top right corner of Figure 2.

To extend relative attention to the local case, we first note that the right block has the same configuration as in the global case (see Figure 1) but much smaller: $(\frac{L}{M})^2$ (where $M$ is the number of blocks, and $N$ be the resulting block length) as opposed to $L^2$. The left block is unmasked with relative indices running from -1 (top right) to $-2N + 1$ (bottom left). Hence, the learned $E^r$ for the local case has shape $(2N - 1, N)$.

Similar to the global case, we first compute $QE^{r\top}$ and then use the following procedure to skew it to have the same indexing as $QK^\top$, as illustrated in Figure 2.

1. Pad a dummy column vector of length $N$ after the rightmost column.

2. Flatten the matrix and then pad with a dummy row of length $N - 1$.

3. Reshape the matrix to have shape $(N + 1, 2N - 1)$.

4. Slice that matrix to retain only the first $N$ rows and last $N$ columns, resulting in a $(N, N)$ matrix.

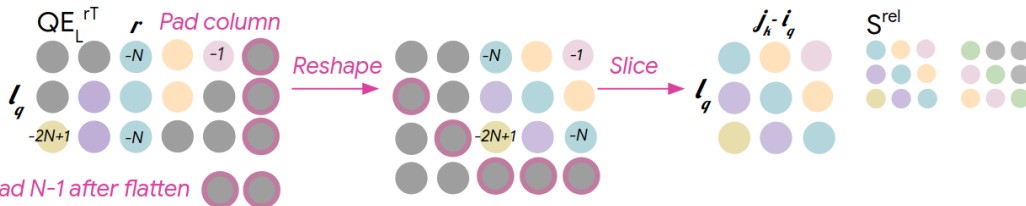

Figure 2: Relative local attention: the thumbnail on the right shows the desired configuration for $S^{rel}$. The "skewing" procedure is shown from left to right.

## 4 EXPERIMENTS

### 4.1 J.S. BACH CHORALES

J.S. Bach Chorales is a canonical dataset used for evaluating generative models for music [3] (e.g., Allan & Williams, 2005; Boulanger-Lewandowski et al., 2012; Liang, 2016; Hadjeres et al., 2016; Huang et al., 2017). It consists of score-based four-part chorales. We first discretize the scores onto a 16th-note grid, and then serialize them by iterating through all the voices within a time step and then advancing time (see A.1 for more details). As there is a direct correspondence between position in sequence and position on the timing/instrument grid in a piece, adding relative position representations could make it easier to learn this grammar. We indeed see relative attention drastically improve negative log-likelihood (NLL) over baseline Transformer (Table 2). This improvement is also reflected in sample quality. The samples now maintain the necessary timing/instrument grid, always advancing four steps before advancing in time. As local timing is maintained, the model is able to capture timing on a more global level, giving rise to regular phrasing, as shown in the right score of Figure 3.

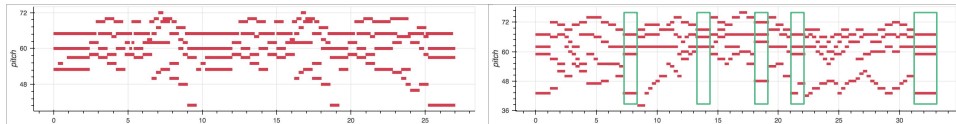

Figure 3: Unconditioned samples from Transformer without (left) and with (right) relative self-attention. Green vertical boxes indicate the endings of (sub)phrases where cadences are held.

In addition to relative attention, we explored enhancing absolute timing through concatenating instead of adding the sinusoids to the input embeddings. This allows the model to more directly learn its absolute positional mapping. This further improves performance for both the baseline and relative transformer (Table 2). We compare against COCONET as it is one of the best-performing models that has also been evaluated on the 16-note grid using the canonical dataset split. To directly compare, we re-evaluated COCONET to obtain note-wise losses on the validation set [4]. For the Transformer models (abbreviated as TF), we implemented our attention mechanisms in the Tensor2Tensor framework (Vaswani et al., 2018). We use 8 heads, and keep the query, key (att) and value hidden size (hs) fixed within a config. We tuned number of layers (L in {4,5,6}), attention hidden size (att in {256, 512}) and pointwise feedforward hidden size (ff in {512, 1024}).

#### 4.1.1 GENERALIZING RELATIVE ATTENTION TO CAPTURE RELATIONAL INFORMATION

A musical event bears multiple attributes, such as timing, pitch, instrument etc. To capture more relational information, we extend relative attention to capture pairwise distances on additional attributes. We learn separate relative embeddings for timing $E^t$ and also pitch $E^p$. $E^t$ has entries corresponding to how many sixteenth notes apart are two positions in time, while $E^p$ embeds the pairwise pitch interval. However this approach is not directly scalable beyond J.S. Bach Chorales because it involves explicitly gathering relative embeddings for $R^t$ and $R^p$, resulting in a memory complexity of $O(L^2D)$ as in Shaw et al. (2018). This is due to relative information being computed based on content as opposed to content-invariant information such as position in sequence. It was sufficient to add these extra relational information to the first layer, perhaps because it is closest to the raw input content. Here, the relative logits are computed from three terms, $S^{rel} = \text{Skew}(QE^r) + Q(R^t + R^p)$ in contrast with other layers that only have one relative term, $S^{rel} = \text{Skew}(QE^r)$.

### 4.2 MAESTRO

MAESTRO (MIDI and Audio Edited for Synchronous TRacks and Organization) (Hawthorne et al., 2019) is a dataset consisting of 172 hours of virtuosic piano performances captured in both MIDI

---

[3]JSB Chorales dataset: `https://github.com/czhuang/JSB-Chorales-dataset`

[4]Some earlier papers report frame-wise losses to compare to models such as RNN-RBM which model "chords". Coconet can be evaluated under note-wise or frame-wise losses.

and audio format [5]. It is curated from the Piano-e-Competitions [6], and proposes a train / valid / test split where the same composition, even if performed by multiple contestants, only appears in one split. The result is 295 / 60 / 75 unique compositions, corresponding to 954 / 105 / 125 performances that last for 140 / 15 / 17 hours and contain 5.06 / 0.54 / 0.57 million notes. The polyphonic MIDI performances contain both expressive dynamics and timing and we serialize them into sequences of event-based tokens as introduced in Oore et al. (2018) (see Section A.2 for more details on the encoding procedure).

We train on random crops of 2048-token sequences and employ two kinds of data augmentation: pitch transpositions uniformly sampled from $\{-3, -2, \ldots, 2, 3\}$ half-steps, and time stretches uniformly sampled from the set $\{0.95, 0.975, 1.0, 1.025, 1.05\}$. For evaluation, we segment each sequence sequentially into 2048 length subsequences and also keep the last subsequences that are of shorter lengths. This results in 1128 and 1183 subsequences in the validation and test set respectively. Each subsequence is then evaluated by running the model forward once with teaching forcing. As the subsequences vary in length, the overall negative loglikelihood (NLL) is averaged entirely on the token level.

Table 2: Note-wise validation NLL (nats/token) on J.S. Bach Chorales where each token is a 16th note. NLL is improved by using the Transformer with our memory-efficient relative global attention formulation, also when including additional positional and relational information.

| Model variation | Validation |
|---|---|
| COCONET (CNN, chronological, 64L, 128 3x3f) | 0.436 |
| COCONET (CNN, orderless, 64L, 128 3x3f) | 0.238 [7] |
| Transformer (TF) baseline (5L, 256hs, 256att, 1024ff, 8h) | 0.417 |
| + concat positional sinusoids | 0.398 |
| + concat positional sinusoids, include instrument labels | 0.370 |
| TF with efficient relative attention (ours) (5L, 512hs, 512att, 512ff, 256r, 8h) | 0.357 |
| + concat positional sinusoids, include instrument labels | 0.347 |
| + relative pitch and time | 0.335 |

Table 3: Validation NLL (nats/token) for Maestro dataset, with event-based representation with lengths $L = 2048$. Training and/or evaluating on different lengths will result in different losses because the amount of context available to the model would be different. The Transformer with our memory-efficient relative attention formulation achieves state-of-the-art results.

| Model variation | Validation | Test |
|---|---|---|
| PERFORMANCE RNN (LSTM) (3L, 1024hs) | 2.094 | — |
| LSTM with attention (3L, 1024hs, 1024att) | 2.066 | — |
| Transformer (TF) baseline (8L, 384hs, 512att, 1024fs, 0.2d) | 1.835 | 1.813 |
| local attention (10L, 512bs, 0.3d) | 1.895 | 1.888 |
| efficient relative local attention (ours) (8L, 512bs, 0.2d) | 1.873 | 1.861 |
| efficient relative global attention (ours) (8L, 1024r, 0.2d) | 1.808 | **1.791** |

We compare our results to PerformanceRNN (LSTM, which first used this dataset) (Oore et al., 2018) and LookBack RNN (LSTM with attention) (Waite, 2016). LookBack RNN uses an input representation that requires monophonic music with barlines which is information that is not present

---

[5]Maestro dataset: urlhttps://magenta.tensorflow.org/datasets/maestro

[6]Piano-e-Competition: `http://www.piano-e-competition.com/`

[7]COCONET is an instance of OrderlessNADE, which approximates a mixture model over orderings where orderings are assumed to be uniformly distributed. Hence, its loss is computed by averaging losses over multiple random orderings. The current row in the table reports this loss. In contrast, the row above corresponds to evaluating Coconet as an autoregressive model under the chronological ordering. Huang et al. (2017) show that sample quality is better when using Gibbs sampling (which uses conditionals from multiple orderings) as opposed to autoregressive generation (which only uses conditionals from one ordering).

in performed polyphonic music data, hence we simply adopt their architecture. Table 3 shows that Transformer-based architectures fits this dataset better than LSTM-based models.

We implemented our attention mechanisms in the Tensor2Tensor framework (Vaswani et al., 2018), and used the default hyperparameters for training, with 0.1 learning rate and early stopping. We compare four architectures, varying on two axes: global versus local, and regular versus relative attention. We found that reducing the query and key channel size (att) to three forth of the hidden size (hs) works well for this dataset and used this setup for all of the models, while tuning on number of layers (L) and dropout rate (d). We use block size (bs) 512 for local attention (Liu et al., 2018; Parmar et al., 2018). For relative global attention, the maximum relative distance to consider is set to half the training sequence length. For relative local attention, it is set to the full memory length which is two blocks long.

Table 3 shows that our memory-efficient relative attention formulations outperform regular attention in both the global and the local case. When looking at the other axes, we see global attention outperforming local attention in both the relative and regular case. Global attention may have the advantage of being able to directly look back for repeating motifs. With a larger dataset, local attention may fare well as it allows for much deeper models and longer sequences, as seen in text and image generation work (Liu et al., 2018; Parmar et al., 2018)). In turn, both domains could benefit from the use of relative local attention.

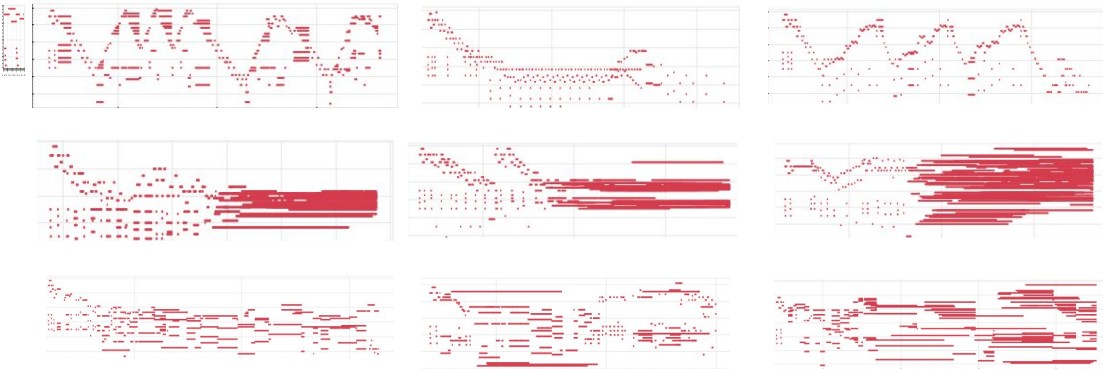

Figure 4: Transformer with relative attention (three random samples on the top row) when given an initial motif (top left) generates continuations with more repeated motifs and structure then baseline Transformer (middle row) and PerformanceRNN (bottom row).

### 4.2.1 QUALITATIVE PRIMING EXPERIMENTS

When primed with an initial motif (Chopin's Étude Op. 10, No. 5) shown in the top left corner of Figure 4, we see the models perform qualitatively differently. Transformer with relative attention elaborates the motif and creates phrases with clear contour which are repeated and varied. Baseline Transformer uses the motif in a more uniform fashion, while LSTM uses the motif initially but soon drifts off to other material. Note that the generated samples are twice as long as the training sequences. Relative attention was able to generalize to lengths longer than trained but baseline Transformer deteriorates beyond its training length. See Appendix C for visualizations of how our Relative Transformer attends to past motifs.

### 4.2.2 HARMONIZATION: CONDITIONING ON MELODY

To explore the sequence-to-sequence setup of Transformers, we experimented with a conditioned generation task where the encoder takes in a given melody and the decoder has to realize the entire performance, i.e. melody plus accompaniment. The melody is encoded as a sequence of tokens as in Waite (2016), quantized to a 100ms grid, while the decoder uses the performance encoding described in Section 3.1 (and further illustrated in A.2). We use relative attention on the decoder side and show in Table 4 that it also improves performance.

Table 4: Relative attention improves conditioned negative logliklihood (NLL) given groundtruth melodies from the validation split of the Maestro dataset.

| Model variation | Validation NLL |
|---|---|
| Transformer baseline | 2.066 |
| Transformer with efficient relative attention (ours) | 1.786 |

### 4.2.3 HUMAN EVALUATIONS

To compare the perceived sample quality of models trained on the Maestro dataset, and their ability to generate a continuation for a priming sequence, we carried out a listening test study comparing the baseline Transformer, our Transformer with relative-attention, PerformanceRNN (LSTM), and the validation set. Participants were presented with two musical excerpts (from two different models that were given the same priming sequence) and asked to rate which one is more musical on a Likert scale. For each model, we generated 10 samples each with a different prime, and compared them to three other models, resulting in 60 pairwise comparisons. Each pair was rated by 3 different participants, yielding a total of 180 comparisons.

Figure 5 shows the number of comparisons in which an excerpt from each model was selected as more musical. The improvement in sample quality from using relative attention over the baseline Transformer model was statistically significant (see Appendix B for the analysis), both in aggregate and between the pair. Even though in aggregate LSTMs performed better in the study than the Transformer, despite having higher perplexity, but when compared against each other head to head, the results were not statistically significant (see Table 5 in Appendix B).

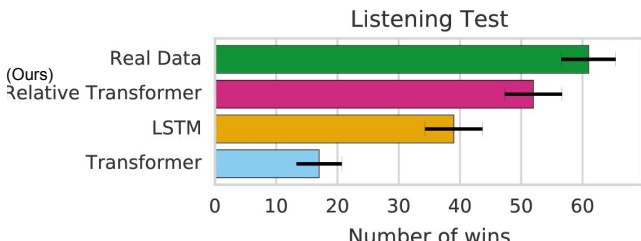

Figure 5: Samples from the Transformer with our efficient relative attention were rated more musical more times than LSTM and baseline Transformer. Error bars show standard deviation of the mean.

## 5 CONCLUSION

In this work we demonstrated that the Transformer equipped with relative attention is very well-suited for generative modeling of symbolic music. The compelling long-term structure in the samples from our model leaves us enthusiastic about this direction of research. Moreover, the ability to expand upon a prime, in particular, suggests potential applications as creative tool.

The significant improvement from relative attention highlights a shortcoming of the original Transformer that might also limit its performance in other domains. Improving the Transformer's ability to capture periodicity at various time scales, for instance, or relations between scalar features akin to pitch could improve time-series models. Our memory-efficient implementation enables the application of relative attention to much longer sequences such as long texts or even audio waveforms, which significantly broadens the range of problems to which it could be applied.

## 6 ACKNOWLEDGEMENT

We thank many colleagues from the Transformer (Vaswani et al., 2017) and Tensor2Tensor (Vaswani et al., 2018) papers for helping us along the way: Lukasz Kaiser, Ryan Sepassi, Niki Parmar and Llion Jones. Many thanks to Magenta and friends for their support throughout and for many insightful

discussions: Jesse Engel, Adam Roberts, Fred Bertsch, Erich Elsen, Sander Dieleman, Sageev Oore, Carey Radebaugh, Natasha Jaques, Daphne Ippolito, Sherol Chan, Vida Vakilotojar, Dustin Tran, Ben Poole and Tim Cooijmans.

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

# A    DOMAIN-SPECIFIC REPRESENTATIONS

Adapting sequence models for music requires making decisions on how to serialize a polyphonic texture. The data type, whether score or performance, makes certain representations more natural for encoding all the information needed while still resulting in reasonable sequence lengths.

## A.1    SERIALIZED INSTRUMENT/TIME GRID (USED FOR THE J.S.BACH CHORALES DATASET)

The first dataset, J.S. Bach Chorales, consists of four-part score-based choral music. The time resolution is sixteenth notes, making it possible to use a serialized grid-like representation. Figure 6 shows how a pianoroll (left) can be represented as a grid (right), following (Huang et al., 2017). The rows show the MIDI pitch number of each of the four voices, from top to bottom being soprano ($S$), alto ($A$), tenor ($T$) and bass ($B$), while the columns is discretized time, advancing in sixteenth notes. Here longer notes such as quarter notes are broken down into multiple repetitions. To serialize the grid into a sequence, we interleave the parts by first iterating through all the voices at time step 1, and then move to the next column, and then iterate again from top to bottom, and so on. The resulting sequence is $S_1 A_1 T_1 B_1 S_2 A_2 T_2 B_2$..., where the subscript gives the time step. After serialization, the most common sequence length is 1024. Each token is represented as onehot in pitch.

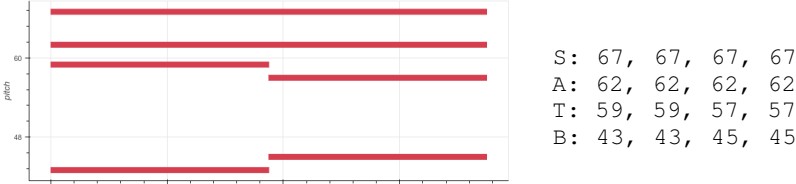

Figure 6: The opening measure of BWV 428 is visualized as a pianoroll (left, where the x-axis is discretized time and y-axis is MIDI pitch number), and encoded in grid representation with sixteenth note resolution (right). The soprano and alto voices have quarter notes at pitches G4 (67) and D4 (62), the tenor has eighth notes at pitches B3 (59) and A3 (57), and the bass has eighth notes at pitches A2 (45) and G2 (43).

## A.2    MIDI-LIKE EVENT-BASED (USED FOR THE MAESTRO DATASET)

The second dataset, Maestro (Hawthorne et al., 2019), consists of polyphonic piano performances with expressive timing and dynamics. The time resolution here is on the millisecond level, so a grid representation would result in sequences that are too long. Instead, the polyphonic performance is serialized into a sequence of one hot encoded events as proposed in Oore et al. (2018).

First, the input MIDI files are preprocessed to extend note durations based on sustain pedal control events. The sustain pedal is considered to be down whenever a sustain control change is encountered with a value $>= 64$; the sustain pedal is then considered up after a control change with a value $< 64$. Within a period where the sustain pedal is down, the duration of each note is extended to either the beginning of the next note of the same pitch or the end of the sustain period, whichever happens first. If the original duration extends beyond the time when the sustain pedal is down, that original duration is used.

Next, the MIDI note events are converted into a sequence from the following set of vocabulary: 128 NOTE_ON events for starting a note of with one of the 128 MIDI pitches, 128 NOTE_OFF events for ending a note with one of the 128 MIDI pitches, 100 TIME_SHIFT events representing forward time shifts in 10ms increments from 10ms to 1s, and 32 SET_VELOCITY events representing the velocity for future NOTE_ON events in the form of the 128 possible MIDI velocities quantized into 32 bins. An example performance encoding is illustrated in Figure 7.

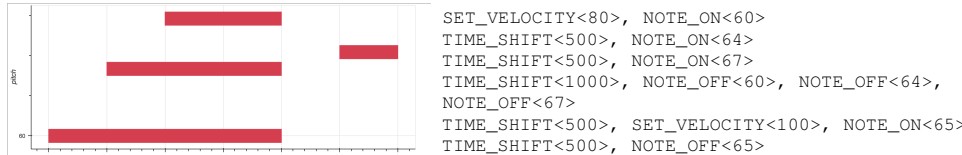

```
SET_VELOCITY<80>, NOTE_ON<60>
TIME_SHIFT<500>, NOTE_ON<64>
TIME_SHIFT<500>, NOTE_ON<67>
TIME_SHIFT<1000>, NOTE_OFF<60>, NOTE_OFF<64>,
NOTE_OFF<67>
TIME_SHIFT<500>, SET_VELOCITY<100>, NOTE_ON<65>
TIME_SHIFT<500>, NOTE_OFF<65>
```

Figure 7: A snippet of a piano performance visualized as a pianoroll (left) and encoded as performance events (right, serialized from left to right and then down the rows). A C Major chord is arpeggiated with the sustain pedal active. At the 2-second mark, the pedal is released, ending all of the notes. At the 3-second mark, an F is played for .5 seconds. The C chord is played at velocity 80 and the F is played at velocity 100.

## B SUPPLEMENT OF LISTENING TEST

### B.1 STUDY PROCEDURE

Participants were presented with two musical excerpts that shared a common priming sequence. For each excerpt, the priming sequence was played, followed by 2.5 seconds of silence, followed by the priming sequence again and a continuation of that sequence. The continuations were either sampled from one of the models or extracted from our validation set. We evaluated all possible pairs in the space of data and model samples, except from the same model. Each continuation had a length of 512 events using the encoding described in Section A.2. This corresponds to the length the models were trained on to remove the deteriorating effect that happens with baseline Transformer when asked to generate beyond the length it was trained on. Participants were asked which excerpt they thought was more musical on a Likert scale of 1 to 5. The pair is laid out left versus right, with 1 indicating the left is much more musical, 2 the left is slightly more musical, 3 being a tie, 4 being the right is slightly more musical, and 5 the right is much more musical. For each model, we generated 10 samples each with a different prime, and compared them to three other models, resulting in 60 pairwise comparisons. Each pair was rated by 3 different participants, yielding a total of 180 comparisons.

### B.2 ANALYSIS

A Kruskal-Wallis H test of the ratings showed that there was a statistically significant difference between the models: $\chi^2(2) = 63.84, p = 8.86\text{e-}14 < 0.01$. Table 5 show a post-hoc analysis on the comparisons within each pair, using the Wilcoxon signed-rank test for matched samples. Table 6 shows a post-hoc analysis of how well each model performed when compared to all pairs, and compares each model's aggregate against each other, using the Mann–Whitney U test for independent samples. We use a Bonferroni correction on both to correct for multiple comparisons. The win and loss counts bucket scores 4, 5 and scores 1, 2 respectively, while the tieing score is 3.

Both within pairs and between aggregates, participants rated samples from our relative Transformer as more musical than the baseline Transformer with $p < 0.01/6$.

For within pairs, we did not observe a consistent statistically significant difference between the other model pairs, baseline transformer versus LSTM and LSTM versus relative Transformer.

When comparing between aggregates, LSTM was overall perceived as more musical than baseline Transformer. Relative Transformer came a bit close to outperforming LSTM with $p = 0.018$. When we listen to the samples from the two, they do sound qualitatively different. Relative Transformer often exhibits much more structure (as shown in Figure 4), but the effects were probably less pronounced in the listening test because we used samples around 10s to 15s, which is half the length of those shown in Figure 4 to prevent the baseline Transformer from deteriorating. This weakens the comparison on long-term structure.

When compared to real music from the validation set, we see that in aggregates, real music was better than LSTM and baseline Transformer. There was no statistical significant difference between real music and relative Transformer. This is probably again due to the samples being too short as real music is definitely still better.

Table 5: A post-hoc comparison of each pair on their pairwise comparisons with each other, using the Wilcoxon signed-rank test for matched samples. $p$ value less than 0.01/6=0.0016 yields a statistically significant difference and is marked by asterisk.

| Pairs | | wins | ties | losses | $p$ value |
|---|---|---|---|---|---|
| Our relative transformer | real music | 11 | 4 | 15 | 0.243 |
| Our relative transformer | Baseline transformer | 23 | 1 | 6 | 0.0006* |
| Our relative transformer | LSTM | 18 | 1 | 11 | 0.204 |
| Baseline transformer | LSTM | 5 | 3 | 22 | 0.006 |
| Baseline transformer | real music | 6 | 0 | 24 | 0.0004* |
| LSTM | real music | 6 | 2 | 22 | 0.0014 |

Table 6: Comparing each pair on their aggregates (comparisons with all models) in (wins, ties, losses), using the Mann–Whitney U test for independent samples.

| Model | | Model | | $p$ value |
|---|---|---|---|---|
| Our relative transformer | (52, 6, 32) | real music | (61, 6, 23) | 0.020 |
| Our relative transformer | (52, 6, 32) | Baseline transformer | (17, 4, 69) | 1.26e-9* |
| Our relative transformer | (52, 6, 32) | LSTM | (39, 6, 45) | 0.018 |
| Baseline transformer | (17, 4, 69) | LSTM | (39, 6, 45) | 3.70e-5* |
| Baseline transformer | (17, 4, 69) | real music | (61, 6, 23) | 6.73e-14* |
| LSTM | (39, 6, 45) | real music | (61, 6, 23) | 4.06e-5* |

## C  VISUALIZING SOFTMAX ATTENTION

One advantage of attention-based models is that we can visualize its attention distribution 3. This gives us a glimpse of how the model might be building up recurring structures and how far it is attending back. The pianorolls in the visualizations below is a sample generated from Transformer with relative attention. Each figure shows a query (the source of all the attention lines) and previous memories being attended to (the notes that are receiving more softmax probabiliy is highlighted in). The coloring of the attention lines correspond to different heads and the width to the weight of the softmax probability.

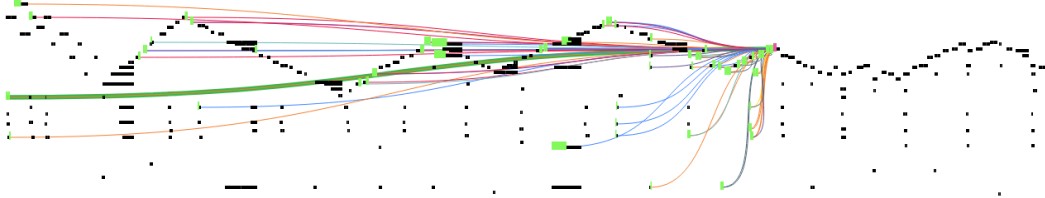

Figure 8: This piece has a recurring triangular contour. The query is at one of the latter peaks and it attends to all of the previous high notes on the peak, all the way to beginning of the piece.

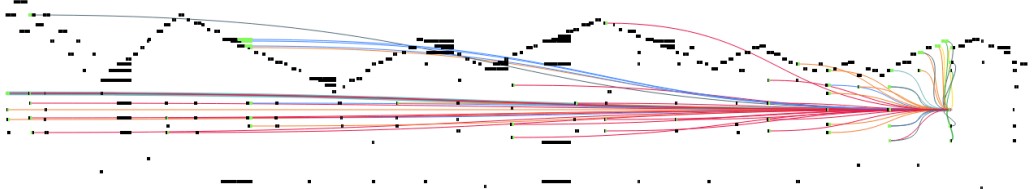

Figure 9: The query a note in the left-hand, and it attends to its immediate past neighbors and mostly to the earlier left hand chords, with most attention lines distributed in the lower half of the pianoroll.

## D    PREVIOUS FIGURES FOR THE "SKEWING" PROCEDURE

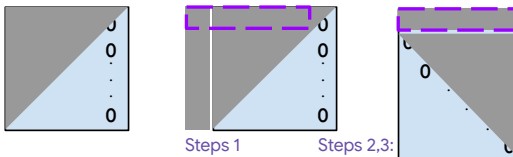

Figure 10: Relative global attention: Steps (from left to right) for "skewing" an absolute-by-relative $(i_q, r)$ indexed matrix into absolute-by-absolute $(i_q, j_k)$. Grey indicates self-attention masks or entries introduced by the skewing procedure. Positions with relative distance zero are marked. Entries outlined by purple are removed in step 3.

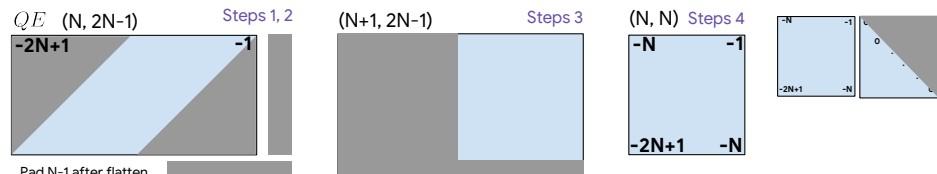

Figure 11: Relative local attention: Steps (from left to right) for "skewing" an $(i_q, r)$ indexed matrix with $2N - 1$ ranged relative indices $r$ into $(i_q, j_k$ indexed. Shapes are indicated above the boxes, while indices in the boxes give relative distances.

