# OpenReview forum: "Music Transformer: Generating Music with Long-Term Structure"
_ICLR.cc/2019/Conference_

### Official Review · AnonReviewer1 · 2018-11-02
**Improved efficiency of transformer on long sequences, but a bit difficult to follow**

**Rating:** 5
**Confidence:** 3

**Review:**

This paper describes a method for improving the (sequence-length) scalability of the Transformer architecture, with applications to modeling long-range interactions in musical sequences. The proposed improvement is applied to both global and local relative attention formulations of self-attention, and consists of a clever re-use (and re-shaping) of intermediate calculations. The result shaves a factor of L (sequence length) from the (relative) memory consumption, facilitating efficient training of long sequences. The method is evaluated on MIDI(-like) data of Bach chorales and piano performances, and compares favorably to prior work in terms of perplexity and a human listener evaluation.

The results in this paper seem promising, though difficult to interpret.  The quantitative evaluation consists of perplexity
scores (Tables 2 and 3), and the qualitative listening study is analyzed by pairwise comparisons between methods. While the proposed method achieves the highest win-rate in the listening study, other results in the study (LSTM vs Transformer) run contrary to the ranking given by the perplexity scores in Table 3. This immediately raises the question of how perceptually relevant the (small) differences in perplexity might be, which in turn clouds the overall interpretation of the results. Of course, perplexity is not the whole story here: the focus of the paper seems to be on efficiency, not necessarily accuracy, but one might expect improved efficiency to afford higher model capacity and improve on accuracy.


The core contributions of this work are described in sections 3.4 and 3.5, and while I get the general flavor of the idea, I find the exposition here both terse and difficult to follow. Figures 1 and 2 should illustrate the core concept, but they lack axis labels (and generally sufficient detail to decode properly), and seem to use the opposite color schemes from each-other to convey the same ideas.  Concrete image maps using real data (internal feature activations) may have been easier to read here, along with an equation that describes how the array indices map after skewing.

The description in 3.4 of the improved memory enhancement is also somewhat difficult to follow.  The claim is a reduction from O(DL^2) to O(DL), but table 1 lists this as O(DL^2) to O(DL + L^2).  In general, I would expect L to dominate D, which still leaves the memory usage in quadratic space, so it's not clear how or why this constitutes an improvement. The improvement due to moving from global to local attention is clear, but this does not appear to be a contribution of this work.

---

> ### Author Response · Authors · 2018-11-27
> **We revised the prose on describing our skewing procedure (sections 3.4 and 3.5) and made new figures for the paper to make the explanations more intuitive.**
>
> Thank you for your review and suggestions.
>
> We first clarify that we are not reducing the memory requirements of the Transformer architecture from Vaswani et al. (2017), which is O(L^2).  Relative attention as proposed by Shaw et al. (2018) involves instantiating an additional intermediate relative embedding that requires O(DL^2).  With our new formulation, we reduce this component to O(DL).  The overall relative attention memory complexity is still O(L^2), but with the added benefit of incorporating relational information which improves perplexity and generation.
>
> Perplexity and listening tests evaluate different objectives.  We do not know if between different model classes, perplexity and listening evaluations correlate monotonically.  However, when comparing baseline Transformer and our relative Transformer, the latter performs better both in perplexity and listening tests.  Figure 4 shows that samples from relative attention exhibit a lot more structure and better generalization (i.e. maintaining coherence over twice the length it was trained on), while both is not true for baseline Transformer.   One can also clearly hear the difference from the music samples that was included in the link below:
>
> https://storage.googleapis.com/music-transformer/index.html
>
> From the link above, you can also hear and contrast unconditioned samples from our relative Transformer and samples taken from prior work (Oore et al., 2018).  We believe you will hear there is a difference.   Before our work, LSTMs were used at time scales of 15s (~ 500 tokens) on the Piano-e-Competition dataset.  Our work shows that Transformers not only model these complex expressive piano performances better, and can also do this at scales of 60s (~2000 tokens) with remarkable long-term coherence.
>
> We have revised sections 3.4 and 3.5 and made new figures (with axes labels) to make the explanations more intuitive.  We agree the previous Figures 1 and 2 were harder to read, even though they did bear the same coloring scheme, with gray indicating positions that were either masked out or padded.  In the new figures we added additional color coding for the different relative distances to make it easier to see the correspondances.  We also added an equation to describe how the array indices map before and after skewing.  Before, we have an absolute-by-relative (i_q, r) indexed matrix, and after skewing we have an absolute-by-absolute (i_q, j_k) indexed matrix, where j_k = r - (L-1) + i_q.

---

> > ### Comment · AnonReviewer1 · 2018-11-28
> > **Clarifications were much appreciated**
> >
> > I've looked over the revised draft, and it is definitely easier to understand now.  Figures 1 and 2 are much clearer now.   Table 1 also puts the algorithmic contributions more into perspective here, as the asymptotic notation for memory complexity does tend to sweep quite a bit under the rug.
> >
> > The examples are indeed compelling, and do demonstrate a qualitative improvement over the prior work (at least in the cases included here).
> >
> > I agree that perplexity and listening tests are different kinds of evaluation, and that one shoudn't necessarily expect monotonic agreement between the two.  However, since perplexity scores are not directly interpretable, and the listening test results can help to anchor the perplexity scores.  Still, I find it difficult to parse how meaningful a difference of 0.023 nats is (for example)  in table 3, or whether the difference of 0.03 in table 2 is indeed "drastic".

---

> > > ### Author Response · Authors · 2018-12-05
> > > **The perplexity improvements are statistically significant.**
> > >
> > > Thank you for looking over our revised draft.  We’re glad the clarifications were helpful.
> > >
> > > The seemingly small numbers in the improvement is because the unit of evaluation is small, being an attribute (such as loudness, pitch etc) of a note, analogous to sub-pixel level autoregressive evaluation.  For Table 3 (Piano-e-Competition), our sequences are of length 2048, a 0.023 nats improvement on the sub-note level is a 47.10 nats on the sequence level.  Similarly for Table 2 (JSB Chorales), the unit is on a discretized grid of 16th notes and sequences are of length 1024.  An improvement of 0.03 nats per token corresponds to a 30.72 nats improvement per sequence.
> > >
> > > We show below a statistical analysis of the results, and we find that the perplexity improvements on both datasets, JSB Chorales and Piano-e-Competition, to be statistically significant.
> > >
> > > First, for Piano-e-Competition the test set results for the last four rows of Table 3 are Transformer baseline 1.852 nats/token, local attention 1.840, relative attention (ours) 1.803, relative local attention (ours) 1.817.
> > >
> > > To compare a pair of models, we perform the post-hoc Wilcoxon signed-rank test, which allows us to determine if there is a statistical difference in the negative loglikelihoods (NLL) under the models.  The Wilcoxon signed-rank test is a standard non-parametric test that tests for paired differences.  In our case, each sequence in the test set is evaluated by a pair of models, and within-pair differences in NLL are calculated to perform the test.
> > >
> > > For the Piano-e-Competition dataset, we report the pairs of model that pertain to our model improvements to strongest baseline, where N = 125, the number of sequences in the test set.  Between local attention (with mean = 1.840 nats/token) and relative local attention (ours) (with mean = 1.817), p-value = 1.69e-19 < 0.01.
> > > Between local attention (with mean=1.840 nats/token) and relative attention (ours) (with mean = 1.803), p-value = 4.00e-21 < 0.01.
> > > The extreme low p-values reflect that the relative improvement between the models are large.  For sanity check, we see that between relative attention (ours) (with mean = 1.803) and relative_local (ours) (with mean = 1.817), which have much closer NLL (improvements under the test set is 0.014 nats/token, the validation set is 0.005 based on Table 3), the difference is still statistically significant, but with a much larger p-value, where p-value=6.88e-05 < 0.01.
> > >
> > > Similarly for the JSB Chorale dataset, we report the test set results for the Transformer baseline and Transformer with relative attention, which are 0.407 nats/token and 0.357 respectively, corresponding to the top rows of the bottom two row groups in Table 2.  We were sadly not able to rerun our best model for this dataset (bottom row of Table 2) due to changing code bases since nearly a year ago, and will fix that for the next version.   For now, we show the statistical test on the Transformer baseline and Transformer with relative attention aforementioned, which shows a test set improvement of 0.05.  The Wilcoxon signed-rank test with N = 77, the number of sequences in the test set, on the paired differences give a p-value of 2.46e-11 < 0.01, showing that the difference between Transformer baseline and Transformer with relative attention is statistically significant.

---

### Official Review · AnonReviewer3 · 2018-11-03
**An application of transformer to music generation**

**Rating:** 4
**Confidence:** 4

**Review:**

In this paper the authors propose an algorithm to reduce the memory
requirements for calculating relative position vectors in a
self-attention (transformer) network, based on the work of [Vaswani et
al., 2017; Shaw et al. 2018]. The authors applied their model to a music
generation task, and evaluated it on two datasets (J.S. Bach Chorales
and Piano-e-Competition). Their model obtained improvements over the
state-of-the-art in the Piano-e-Competition set in terms of
log-likelihoods. Additionally, they performed human evaluation on the
Piano-e-Competition set showing preference of the participants for their
method over the state-of-the-art.

The application of the transformer network seems suitable for the task,
and the authors fairly justify their motivations and choices. They show
improvements over the-state-of-the-art for one data-set and explained
their results. They also show an interesting application of
sequence-to-sequence models for generating complete pieces of music
based on a given melody.

My main concern is the novelty of the paper. The authors use the model
proposed by [Shaw et al. 2018] with an additional modification to manage
very long sequences proposed by [Liu et al., 2018; Parmar et al., 2018],
(chunking the input sequences in non-overlaping blocks and calculating
attention only on the current and the previous blocks). Their main
contribution is to reduce the memory requirement for matrix operations
for calculating the relative position vectors of the self-attention
function, which was sub-optimal in [Shaw et al. 2018]. The memory
reduction is from O(L^2D+L^2) to O(LD+L^2). I would qualify this as an
optimization in the implementation of the existing method rather than a
new approach.

---

> ### Author Response · Authors · 2018-11-27
> **We introduce novel use of the Transformer for several musical tasks, and provide state-of-the-art empirical results.**
>
> Thank you for your review.
>
> We would like to point out that the major contribution of this paper is empirical, where we are the first to successfully adapt a self-attention based model to generate minute-long (~2000 tokens) sequences of music that sound realistic to human listeners. This is a very difficult problem because of the complicated grammar of music.  In particular, we are modeling both music composition and the performance of it at once, which involves modeling relationships simultaneously at timescales ranging 4 orders of magnitude, from 10 milliseconds to 100s.  Before our work, the state-of-the-art was to use LSTMs to generate 15s of music (Oore et al., 2018).  With our results, we hope that the music community will adopt relative self-attention for modeling music.
>
> We have shown novel use of the Transformer on a range of musical tasks, which yielded novel findings that are useful beyond the music domain.  For example, we see for conditioned generation, when given an initial motif, relative transformer is able to reuse it in a coherent fashion to generate continuations.  This was not possible with LSTMs because it favours recency and soon forgets the initial motifs.  In contrast, transformers can directly look back to “copy” past motifs, however without relative attention the inductive bias was not strong enough for this to happen over longer timescales.  Furthermore, relative transformer was able to generalize beyond the lengths it was trained on.  This was not possible for baseline Transformer.  Both phenomena are shown in Figure 4 and can also be heard clearly in the accompanying audio clips at https://storage.googleapis.com/music-transformer/index.html.
>
> We also show a novel formulation of the harmonization task, given a melody generate an accompaniment, as a seq2seq problem.  The benefit is that even though the accompaniment can only see its own past, it always has full access to the entire melody, allowing it to attend to and account for the future directly.  From the link above, you can hear the model’s accompaniment to “twinkle twinkle little star”.  The accompanying styles of piano playing differs across samples, yet maintains consistency within.
>
> In additional to our domain contributions, we hope that the reviewer will find our algorithmic contributions that reduce the memory footprint from L^2D (8.5 GB per attention layer) to LD (4.2 MB per attention layer) to be useful. This is critical for applying relative transformer to other tasks with long sequences, such as autoregressive models of images that use self-attention (Parmal et al., 2018) and for modeling long sequences in dialogue and summarization.

---

> > ### Comment · AnonReviewer3 · 2018-12-06
> > **Not enough  contribution for a machine learning conference like ICLR.**
> >
> > After reading the revised paper and the responses, I agree that the improvement in the Piano-e-Competition set over the presented baselines also constitutes a contribution of the paper in the music generation domain. The presented samples are compelling, and show that the model actually can produce longer sequences than the compared baselines.
> >
> > While the human evaluation clearly show preference towards the proposed model in comparison with the baselines (two transformer, and one LSTM-based), the same improvement is not so clear in the automatic evaluation with NLL. The authors could include significance test for their results in tables 2 and 3.
> >
> > Considering the above comments, I still have reservations on the novelty and contributions of the paper for a conference like ICLR. I would surely accept this paper in a conference of computational music domain or to ICLR workshop.

---

> > > ### Author Response · Authors · 2018-12-06
> > > **Our improvements in NLL are statistically significant.  Please also consider the impact of this work on modeling long sequences and long-term structure, which are important problems in ML.**
> > >
> > > Thank you for your response.   Another reviewer also recently asked about significance test for results in Tables 2 and 3.  Since we responded under their comment, we are also copying it here below.  The NLL improvements on both the JSB Chorales and Piano-e-Competition are statistically significant.
> > >
> > > We also hope you consider the impact of this work on modeling long sequences, allowing us to move from studying sequences of length 650 to 3500.
> > >
> > > Furthermore,  music is an important domain for studying long-term dependencies, as it involves repetition and self-reference on multiple timescales.  Generative modeling in music is a complex real-world problem that extends traditional synthetic tasks such as copying memory (Hochreiter and Schmidhuber 1997), along with other canonical tasks in text, images, speech and video.  As music possesses a different kind of of long-term structure, as we move forward in research on long-term dependencies, studying a wider range of tasks will allow us to develop better techniques.
> > >
> > >
> > > Below we are quoting our earlier responses on our statistical test on NLL improvements.
> > > """
> > > The seemingly small numbers in the improvement is because the unit of evaluation is small, being an attribute (such as loudness, pitch etc) of a note, analogous to sub-pixel level autoregressive evaluation.  For Table 3 (Piano-e-Competition), our sequences are of length 2048, a 0.023 nats improvement on the sub-note level is a 47.10 nats on the sequence level.  Similarly for Table 2 (JSB Chorales), the unit is on a discretized grid of 16th notes and sequences are of length 1024.  An improvement of 0.03 nats per token corresponds to a 30.72 nats improvement per sequence.
> > >
> > > We show below a statistical analysis of the results, and we find that the perplexity improvements on both datasets, JSB Chorales and Piano-e-Competition, to be statistically significant.
> > >
> > > First, for Piano-e-Competition the test set results for the last four rows of Table 3 are Transformer baseline 1.852 nats/token, local attention 1.840, relative attention (ours) 1.803, relative local attention (ours) 1.817.
> > >
> > > To compare a pair of models, we perform the post-hoc Wilcoxon signed-rank test, which allows us to determine if there is a statistical difference in the negative loglikelihoods (NLL) under the models.  The Wilcoxon signed-rank test is a standard non-parametric test that tests for paired differences.  In our case, each sequence in the test set is evaluated by a pair of models, and within-pair differences in NLL are calculated to perform the test.
> > >
> > > For the Piano-e-Competition dataset, we report the pairs of model that pertain to our model improvements to strongest baseline, where N = 125, the number of sequences in the test set.  Between local attention (with mean = 1.840 nats/token) and relative local attention (ours) (with mean = 1.817), p-value = 1.69e-19 < 0.01.
> > > Between local attention (with mean=1.840 nats/token) and relative attention (ours) (with mean = 1.803), p-value = 4.00e-21 < 0.01.
> > > The extreme low p-values reflect that the relative improvement between the models are large.  For sanity check, we see that between relative attention (ours) (with mean = 1.803) and relative_local (ours) (with mean = 1.817), which have much closer NLL (improvements under the test set is 0.014 nats/token, the validation set is 0.005 based on Table 3), the difference is still statistically significant, but with a much larger p-value, where p-value=6.88e-05 < 0.01.
> > >
> > > Similarly for the JSB Chorale dataset, we report the test set results for the Transformer baseline and Transformer with relative attention, which are 0.407 nats/token and 0.357 respectively, corresponding to the top rows of the bottom two row groups in Table 2.  We were sadly not able to rerun our best model for this dataset (bottom row of Table 2) due to changing code bases since nearly a year ago, and will fix that for the next version.   For now, we show the statistical test on the Transformer baseline and Transformer with relative attention aforementioned, which shows a test set improvement of 0.05.  The Wilcoxon signed-rank test with N = 77, the number of sequences in the test set, on the paired differences give a p-value of 2.46e-11 < 0.01, showing that the difference between Transformer baseline and Transformer with relative attention is statistically significant.
> > > """

---

### Official Review · AnonReviewer4 · 2018-11-10
**Cool idea, memory usage could be analysed deeper**

**Rating:** 6
**Confidence:** 4

**Review:**

The authors address the problem raised by applying a fully attentional network (FAN) to model music.
They argue clearly for the need of relational positional embedding in that problem (instead of absolute positional as in vanilla FAN), and highlight the quadratic memory footprint of the current solution (Shaw et al. 2018).

The main contribution of the paper is a solution to this, consisting in a smart idea (sect 3.4.1 and 3.4.2) which allows them to compute relative embeddings without quadratic overhead.
The model performs indeed better than Shaw et al.'s on the single data-set they compared both. On the other one, the argument is that Shaw et al. 2018 cannot be applied because the sequences are too long.

I have two concerns with the paper:
	1/ it is very hard to read at times. In particular, the main contribution took me several passes the understand. I list below a few recommendations for improvement
	2/ the main argument is that the model requires less memory and is faster. However, the only empirical evidence in that direction is given in the introduction (Sect 1.1., second paragraph).
		The following points remain unclear to me:
			a) why can't the Relative Transformer be applied to Piano-e composition. What is the maximal length that is possible?
			b) how much faster / less memory is the relative music transformers? The only data-point is in Sect 1.1., which seems indeed impressive (but then one wonders why this is not exploited further). A deeper analysis of the comparative memory footprint would greatly strengthen the paper in my opinion.

Why "music" relative transformers? Nothing in the model restrict it to that use case. The use of FAN over audio has been explored with limited success, one of the reasons being that - similarly to this use-case here - audio sequences tend to be longer than text.

minor comments:
	- abstract, ln9: there seems to be a verb missing
	- p1,ln-2: "dramatic" improvements seems to be exaggerated
	- p2,ln11: "too long". too long for what?
	- p4,ln15: (Table 1). is one sentence by itself. Also, a clear explanation of that table is missing
	- p5,item 2: an explanation in formula would be helpful for those not familiar with reshaping
	- Fig3: it seems very anecdotical. Similar green bloxes might be placed on the left plot
	- sect4.1.1,ln3. that sentence does not parse
	- Table 2: what is cpsi?
	- $l$ is nicer formatted as $\ell$
	- care should be taken to render the Figures more readable (notably the quality of Fig 4, and labels of Fig 7)
	- footnotes in Figures are not displayed (Table 2 and 4)
	- the description of the human evaluation leaves some open questions. I could not come up with 180 ratings (shouldn't it be 180 * 3 ratings?). Also, at least the values of Relative Transformer vs other 3 models should be shown (or all 6 comparisons). Here you call "relative transformer" your model, previously you used that term to refer to (Shaw et al. 2018).
		when reporting statistical significance, there are some omissions which should be clarified.
	- (Shaw et al. 2018) has been published at NAACL. For such an important citation, you should update the reference from the arxiv version.

---

> ### Author Response · Authors · 2018-11-27
> **Our main contribution is in music generation, and we have added a table in the paper to provide a deeper analysis of the memory footprint.**
>
> Thank you for your detailed review.
>
> The main contribution of our paper is in generating music with long-term structure, at the timescale of 60s.  We have modified our title and also contribution section in the paper to highlight this point.  Prior work only aimed to generate 15s of expressive piano music, which is ~500 tokens (Oore et al., 2018).  Even with a large GPU with 16GB of memory, the relative attention formulation by Shaw et al. (2018) can only fit ~650 tokens.  With our memory-efficient formulation, we can fit 5x (~3500 tokens), and hence allowing us to experiment with generating minute-long music (~ 2000 token per minute).
>
> We have added a table in the paper to show the memory requirements for the maximal length under each of the formulations, and we also summarize it here.  Assuming hidden size D=512, the memory requirements at 650 tokens is 865 MB for prior method of complexity O(L^2D) and 1.3MB for ours with complexity O(LD).  At 3500 tokens, prior is 25GB, ours is 7.2MB.
>
> Even though the time complexity of both methods are the same O(L^2D), in practice because prior work requires more memory, at length 650 our method is 6x faster.  As memory grows quadratically with length, for longer sequences such as 3500 the difference would be even greater if the comparison was possible.
>
> We titled the paper music transformer because we are the first to apply transformers to music and with our reformulation we were able to use it to significantly advance the state-of-the-art in generating long-scale music.  We also casted music harmonization as a seq2seq task, leveraging the encoder-decoder structure of transformers.  You can hear samples here: https://storage.googleapis.com/music-transformer/index.html.  We agree our contribution can also be useful for other domains that have long sequences and carry long-range dependencies.
>
> Clarifications on the listening test:
> We generated 10 samples for each model, and each model was compared to 3 other models, hence each model was involved in 30 pairwise comparisons.  In other words, since there are 4 models, hence 6 pairs, each pair of models comparing their 10 samples, yielding 60 pairwise comparisons.  Each was rated by 3 different participants, resulting in a total of 180 pairwise comparisons.  In the appendix, we have added the win, tie, loss counts for all 6 pairs, and the details of the statistical tests.
>
> In the paper, whenever we refer to “relative transformer” we have added clarification whether it is our formulation or Shaw et al.’s (2018).   Thank you for catching our typos and suggesting better ways of formatting.  We have updated them accordingly.

---

> > ### Author Response · Authors · 2018-12-06
> > **We have included additional statistical test showing that our NLL improvements are statistically significant.**
> >
> > We had previously answered reviewer 1 on the statistical significance of our NLL improvements, we wanted to point you to their thread in case you had similar questions.  Our analysis showed that both NLL improvements on JSB Chorales and Piano-e-Competition are statistically significant.
> >
> > With our previous and current comments, we hope we have addressed your concerns.  Could you give an updated impression of the paper?

---

### Official Review · AnonReviewer2 · 2018-11-11
**Implementation trick to reduce memory footprint of transformer / experiments on music generation**

**Rating:** 7
**Confidence:** 3

**Review:**

This paper presents an implementation trick to reduce the memory footprint of relative attention within a transformer network. Specifically, the paper points out redudant computation and storage in the traditional implementation and re-orders matrix operations and indexing schemes to optimize. As an appllication, the paper applies the new implementation to music modeling and generation. By reducing the memory footprint, the paper is able to train the transformer with relative attention on longer musical sequences and larger corpora. The experimental results are compelling -- the transformer with relative attention outperforms baselines in terms of perplexity on development data (though test performance is not reported) and by manual evaluation in a user study.

Overall, I'm uncomfortable accepting this paper in its current form because I'm not sure it constitutes a large enough unit of novel work. The novelty here, as far as I can tell, is essentially an implementation trick rather than an algorithm or model. Transformer networks have been applied to music in past work -- the only difference here is that because of the superior implementation the model can be trained from larger musical sequences. All that said, I do think the proposed implementation is useful and that the experimental results are compelling. Clearly, when trained from sufficient data, transformer networks have something to offer that is different from past techniques.

---

> ### Author Response · Authors · 2018-11-27
> **Our work is the first application of Transformers to music generation, with significant advancement to state-of-the-art, also works well on small datasets**
>
> Thank you for reviewing our paper.
>
> As far as we know, our work is the first to apply the Transformer architecture to music, and to model complex music sequences at lengths much longer then previously attempted.  Do you have a reference to the previous application of Transformer to music?
>
> Before our work, LSTMs were used at time scales of 15s (~ 500 tokens) on the Piano-e-Competition dataset (Oore et al., 2018).  Our work shows that Transformers not only model these complex expressive piano performances better, and can also do this at scales of 60s (~2000 tokens) with remarkable long-term coherence.  We invite you to listen to our samples at https://storage.googleapis.com/music-transformer/index.html to see if you agree.  We have also included samples from prior work (Oore et al., 2018) for direct comparison.  Our use of Transformer for music is not only novel but demonstrates a significant advance in the state-of-the-art for music generation.  To achieve this, we had to develop a new algorithm that significantly lowers the space complexity of previous work on relative attention (from x to y) while keeping the computational complexity the same.
>
> It seems the review was cut off at the end, hinting at Transformers requiring larger datasets.
>
> Our work also shows that with relative attention, Transformers can perform extremely well on small datasets such as JSB Chorales, which consists of only 382 pieces, a total of 370 thousand tokens at the 16th note resolution, with an average length of about a thousand per piece.  Without relative attention, the Transformer did not have the right inductive bias to capture longer term structure even though it has the capacity to, and without our work one may suspect that Transformers do not work well for music (which we suspected too initially!).

---

> > ### Author Response · Authors · 2018-12-06
> > **We have included additional statistical test showing that our NLL improvements are statistically significant.**
> >
> > We had previously answered reviewer 1 on the statistical significance of our NLL improvements, we wanted to point you to their thread in case you had similar questions.  Our analysis showed that both NLL improvements on JSB Chorales and Piano-e-Competition are statistically significant.
> >
> > With our previous and current comments, we hope we have addressed your concerns.  Could you give an updated impression of the paper?

---

> > > ### Comment · AnonReviewer2 · 2018-12-12
> > > **First application of transformers to music generation**
> > >
> > > Apologies for my mistake about prior work on applying transformer networks to music: while reading other papers on music generation, I had encountered a few citations of a 2018 paper that directly applied transformer networks to music generation. After going back and inspecting, I found that the paper being cited was in fact the arxiv version of your paper, effectively blowing my mind!
> > >
> > > This changes my opinion. Originally I felt that even as an application paper, the technical novelty was thin since transformers had been applied to music in the past. But given that these results are in fact the first on applying transformers to music, I think they do make sense at ICLR. I have changed my rating accordingly.
> > >
> > > Further, thank you for your diplomatic response!

---

### Author Response · Authors · 2018-11-27
**To all reviewers: Please listen to a new Jazz sample and revisit the previous samples.  We have also revised the title of our paper to be "Music Transformer: Generating music with long-term structure" to highlight our domain contributions.**

Many of the reviewers seemed to think that our main contribution was a more memory-efficient implementation of relative attention. We want to emphasize that this is primarily an application paper; our main contribution is in advancing the state-of-the-art in generative modeling of music, specifically sequences that capture at once a musical composition and an expressive performance of that composition on the piano.  This required getting some details right with the Transformer architecture, which motivated us to explore a more memory-efficient formulation of relative attention.  According to the ICLR call for papers, the conference explicitly cites “applications in vision, audio, speech, natural language processing, robotics, neuroscience, computational biology, or any other field” as a relevant topic, so we feel that ICLR is an appropriate venue for this work.

In addition to looking at perplexity and human eval scores, we urge you to put on headphones and listen to the samples.  We realize the genre of virtuosic classical piano may not be the easiest to resonate with.  We have added a Jazz sample to https://storage.googleapis.com/music-transformer/index.html trained on additional performances.  We have also included the samples posted by prior work (Oore et al., 2018) for direct comparison.  We believe our samples represent a significant advance in quality especially with respect to long-term coherence.

---

### Meta-Review · Area_Chair1 · 2018-12-13
**successful adaptation of transformer networks to generating long coherent music sequences**

**Confidence:** 4
**Recommendation:** Accept (Poster)

**Metareview:**

1. Describe the strengths of the paper.  As pointed out by the reviewers and based on your expert opinion.

- improvements to a transformer model originally designed for machine translation
- application of this model to a different task: music generation
- compelling generated samples and user study.

2. Describe the weaknesses of the paper. As pointed out by the reviewers and based on your expert opinion. Be sure to indicate which weaknesses are seen as salient for the decision (i.e., potential critical flaws), as opposed to weaknesses that the authors can likely fix in a revision.

- lack of clarity at times (much improved in the revised version)

3. Discuss any major points of contention. As raised by the authors or reviewers in the discussion, and how these might have influenced the decision. If the authors provide a rebuttal to a potential reviewer concern, it’s a good idea to acknowledge this and note whether it influenced the final decision or not. This makes sure that author responses are addressed adequately.

The main contention was novelty. Some reviewers felt that adapting an existing transformer model to music generation and achieving SOTA results and minute-long music sequences was not sufficient novelty. The final decision aligns with the reviewers who felt that the novelty was sufficient.

4. If consensus was reached, say so. Otherwise, explain what the source of reviewer disagreement was and why the decision on the paper aligns with one set of reviewers or another.

A consensus was not reached. The final decision is aligned with the positive reviews for the reason mentioned above.